# Vitamin A Rich Diet Diminishes Early Urothelial Carcinogenesis by Altering Retinoic Acid Signaling

**DOI:** 10.3390/cancers12071712

**Published:** 2020-06-28

**Authors:** Daša Zupančič, Jelena Korać-Prlić, Mateja Erdani Kreft, Lucija Franković, Katarina Vilović, Jera Jeruc, Rok Romih, Janoš Terzić

**Affiliations:** 1Institute of Cell Biology, Faculty of Medicine, University of Ljubljana,1000 Ljubljana, Slovenia; dasa.zupancic@mf.uni-lj.si (D.Z.); mateja.erdani@mf.uni-lj.si (M.E.K.); 2Laboratory for Cancer Research, School of Medicine, University of Split, 21000 Split, Croatia; jkorac@mefst.hr (J.K.-P.); lucija.frankovic@mefst.hr (L.F.); 3Department of Pathology, University Hospital of Split, 21000 Split, Croatia; katarina.vilovic@mefst.hr; 4Institute of Pathology, Faculty of Medicine, University of Ljubljana, 1000 Ljubljana, Slovenia; jera.jeruc@mf.uni-lj.si

**Keywords:** urinary bladder, early carcinogenesis, vitamin A, retinoic acid signaling, apoptosis

## Abstract

Urinary bladder cancer is one of the leading malignancies worldwide, with the highest recurrence rates. A diet rich in vitamin A has proven to lower the risk of cancer, yet the molecular mechanisms underlying this effect are unknown. We found that vitamin A decreased urothelial atypia and apoptosis during early bladder carcinogenesis induced by *N*-butyl-*N*-(4-hydroxybutyl) nitrosamine (BBN). Vitamin A did not alter urothelial cell desquamation, differentiation, or proliferation rate. Genes like *Wnt5a*, involved in retinoic acid signaling, and transcription factors *Pparg*, *Ppara*, *Rxra*, and *Hoxa5* were downregulated, while *Sox9* and *Stra6* were upregulated in early urothelial carcinogenesis. When a vitamin A rich diet was provided during BBN treatment, none of these genes was up- or downregulated; only *Lrat* and *Neurod1* were upregulated. The lecithin retinol acyltransferase (LRAT) enzyme that produces all-trans retinyl esters was translocated from the cytoplasm to the nuclei in urothelial cells as a consequence of BBN treatment regardless of vitamin A rich diet. A vitamin A-rich diet altered retinoic acid signaling, decreased atypia and apoptosis of urothelial cells, and consequently diminished early urothelial carcinogenesis.

## 1. Introduction

Urinary bladder carcinoma is the tenth most common malignancy affecting both genders worldwide with high recurrence rate [1,2,3]. Most bladder cancers originate from the urothelium, which forms a blood-urine permeability barrier that protects tissue from carcinogens in the urine [4,5]. The main structural basis for this barrier is urothelial plaques composed of four major integral membrane protein uroplakins, which cover almost the entire apical plasma membrane of terminally differentiated superficial umbrella cells [6,7,8,9]. Earlier studies showed that uroplakin expression and plaque formation are disturbed in urothelial cancers, causing decreased urothelial differentiation [10,11,12,13,14,15,16]. Cancers are usually accompanied by increased proliferation and decreased apoptosis. On the contrary, in early carcinogenesis, apoptotic cell death is increased in the urinary bladder, liver, and thyroid [17,18].

It is known that retinol and retinoic acid (RA) signaling pathways are involved in cellular proliferation, differentiation, apoptosis, and cancer progression [19,20]. An epidemiological study revealed that high vitamin A intake is associated with a lower risk of bladder cancer in humans [21]. Additionally, vitamin A serum levels have been reported to be significantly lower in patients with bladder cancer when compared to controls [22]. Retinol, a form of dietary vitamin A, is delivered to tissues and organs by retinol binding protein (RBP). RBP binds to transmembrane protein stimulated by retinoic acid 6 (STRA6), which facilitates the uptake of retinol by the cell [23,24]. Once inside the cell, aldehyde dehydrogenase (ALDH) can metabolize retinol into active RA [25,26]. RA signals through two classes of nuclear receptors, RAR and RXR, which activate the expression of intermediate transcription factors including FOXA1 and KLF4 [27,28,29,30,31]. On the other hand, lecithin retinol acyltransferase (LRAT) transforms retinol to inactive retinyl esters [32,33,34]. An inverse correlation between LRAT and tumor stage was demonstrated in bladder cancer [35]. STRA6 is known to regulate the homeostasis of adult urothelium as well as urothelial development and regeneration [36,37]. These show that RA signaling is important for urothelial homeostasis as well as tumor development, yet it remains unclear how RA signaling changes in early bladder carcinogenesis and what the effects of high dietary vitamin A on urothelial cells are. 

Here we used *N*-butyl-*N*-(4-hydroxybutyl) nitrosamine (BBN) to induce early bladder carcinogenesis in mice, which is similar to human bladder carcinogenesis in its morphological, biological, and molecular features [38,39,40,41,42]. The amount of vitamin A in the diet was chosen so that an elevated level of the vitamin in the serum of the animals would not be toxic. 

We demonstrate that BBN treatment changes RA signaling and that a vitamin A rich diet diminishes early bladder carcinogenesis by decreasing urothelial atypia and reducing the apoptosis of urothelial cells. Our results suggest that the protective effects of vitamin A are due to increased *Lrat* and neurogenic differentiation 1 (*Neurod1*) expression and LRAT translocation from the cytoplasm of urothelial cells to their nuclei. Increased expression of *Stra6* is observed upon BBN treatment, while a vitamin A rich diet returns *Stra6* to normal levels.

## 2. Results

### 2.1. Vitamin A Protects Against BBN-Induced Urothelial Atypia

We first confirmed that mice on a vitamin A rich diet (VitA and BBN + VitA groups) had higher concentrations of vitamin A in the serum than mice on a standard diet (NT and BBN groups) (Figure 1B). In the NT and VitA groups, urothelium was composed of basal, intermediate, and superficial umbrella cells (Figure 1C,D). In the BBN and BBN + VitA groups, BBN-induced histological changes such as desquamation of urothelium (Figure 1G) and urothelial atypia (Figure 1H) were observed. There were no differences in the extent of desquamation between the two groups (Figure 1G). Urothelial atypia was characterized by the loss of cell polarity, absence of umbrella cells, nuclear crowding, and occasional large pleomorphic and hyperchromatic nuclei with conspicuous nucleoli and irregular contours (Figure 1E,F). In the BBN group, 69% of mice had widespread atypia, while in the BBN + VitA group this was observed in only 52% of mice (Figure 1H). Additionally, 19% of mice in the BBN + VitA group showed normal urothelium with focal atypia. On the other hand, in the BBN group all mice exhibited at least moderate atypia (Figure 1H). The percentage of mice with moderate atypia was comparable in both groups (Figure 1H). Taken together, it can be seen that a vitamin A rich diet diminishes early bladder carcinogenesis progression by decreasing BBN-induced urothelial atypia.

### 2.2. Effects of Vitamin A-Rich Diet and BBN Treatment on Urothelial Cell Differentiation

Keratin 20 (KRT20), uroplakins, and rigid-looking (scalloped) apical surface confirmed terminal differentiation of urothelium in the NT and VitA groups (Figure 2A,B,G,H,K,L). KRT20 revealed two characteristic patterns: (i) Individual superficial urothelial cells with positive labelling in the apical cytoplasm (Figure 2C–F,I,J); and (ii) positive labelling in the cytoplasm of superficial, intermediate, and basal cells (Figure 2D,F) in the BBN and BBN + VitA groups. Uroplakins showed individual superficial cells with positive reaction in the apical cytoplasm in the BBN and BBN + VitA groups (Figure 2I,J). The apical plasma membrane of NT and VitA groups exhibited urothelial plaques characteristic of terminally differentiated cells (Figure 2K,L), while the apical surface of BBN and BBN + VitA groups was covered by ropy ridges and microvilli, which are characteristic of partially differentiated urothelial cells (Figure 2M,N). Quantitative evaluation of these three differentiation markers confirmed a lower differentiation stage of urothelial cells in BBN and BBN + VitA groups in comparison to NT and VitA groups (Figure 2O–Q). There was no change in the differentiation stage between BBN and BBN + VitA groups.

Interestingly, qPCR revealed that the mRNA levels of *Upk1a* and *Upk1b* increased after a vitamin A rich diet (VitA group compared to NT group), while other mRNA levels of UPs (*Upk2*, *Upk3a*, and *Upk3b*) were unaltered (Figure 3). All five uroplakins (*Upk1a*, *Upk1b*, *Upk2*, *Upk3a*, and *Upk3b*) were decreased following BBN treatment (BBN and BBN + VitA groups) (Figure 3). Taken together, it can be seen that although urothelial cell differentiation is not altered by a vitamin A rich diet, it is decreased because of BBN treatment. 

### 2.3. A Vitamin A Rich Diet Decreases Apoptosis and Preserves Proliferation of Urothelial Cells During Early Bladder Carcinogenesis

Proliferating cells expressing Ki-67 were rare in the normal urothelium of NT and VitA groups, with a proliferation index less than 0.8%, which is in agreement with previously published data [30,43]. On the other hand, in the BBN and BBN + VitA groups the average proliferation index was 11.5 ± 0.9% and 11.7 ± 1.0%, respectively (Figure 4C). Most Ki-67 positive cells were localized in the basal layer of the urothelium (Figure 4A). There was no significant difference in proliferation indices of BBN and BBN + VitA groups. Thus, a vitamin A rich diet had no effect on the proliferation of urothelial cells during early bladder carcinogenesis.

On the contrary, the apoptotic index, as determined by active caspase 3 immunolabeling (Figure 4B), was significantly higher in the BBN group than in the BBN + VitA group (Figure 4D). In the NT and VitA groups, the apoptotic index was zero (Figure 4D). We therefore suggest that the protective role of vitamin A in early bladder carcinogenesis is in part the result of decreased apoptosis of urothelial cells (Figure 4D).

### 2.4. Retinoic Acid (RA) Signaling Is Altered during Early Bladder Carcinogenesis, and a Diet Rich in Vitamin A Reduces These Alterations 

First, changes in the expression of genes involved in RA signaling due to BBN treatment were tested. When we compared BBN and NT group, we showed 9 up- and 15 downregulated genes in the BBN group with a high significance level (Chi square = 19.461, *p* = 0.00001) (Table 1, Figure 5A,B). Therefore, we conclude that early bladder carcinogenesis induced by BBN modifies the RA signaling pathway, which might play a crucial role in early malignant transformation of urothelial cells.

Second, we analyzed changes in the expression of genes involved in RA signaling in BBN + VitA and NT group. Here, only *Lrat* was significantly upregulated and only *Foxa1* and *Msx2* were downregulated with a high significance level (Chi square = 19.461, *p* = 0.00001) (Table 1, Figure 5A,C). Therefore, we assume that a vitamin A rich diet decreased the effect of BBN treatment with respect to altered gene expression profile.

Third, we compared the VitA and NT groups. Our results show that a vitamin A rich diet induced an increase in the expression level of *Cyp26b1* and *Lrat* (Figure 6A,B). All other genes were not significantly changed, implying that a vitamin A rich diet does not affect RA signaling in the normal urothelium.

Fourth, we compared BBN + VitA and BBN groups and found differences between them. A vitamin A rich diet together with BBN treatment increased the expression of *Neurod1* and *Lrat* compared to BBN treatment alone (Figure 6A,C). Thus, we suggest that a vitamin A rich diet counteracted the effect of BBN.

### 2.5. LRAT Localization in Urothelial Cells Changes during Early Carcinogenesis Induced by BBN, While STRA6 Localization Is Unaltered

LRAT and STRA6 were present in all urothelial layers in all groups (Figure 7A–H), and were localized in the cytoplasm of urothelial cells in the NT and VitA groups (Figure 7A,B,E,F). In the BBN and BBN + VitA groups, STRA6 localization was unaltered (Figure 7C,D), while LRAT was observed predominantly in the nuclei of urothelial cells in both groups (Figure 7G,H). There was no difference in the localization of LRAT between the BBN and BBN + VitA groups. These results suggest that in normal urothelium, LRAT is localized in the cytoplasm of the urothelial cells, and during early carcinogenesis of the bladder, LRAT is predominantly transported into the nuclei of urothelial cells.

## 3. Discussion

Several studies reported that vitamin A and its derivatives inhibit bladder cancer progression [21,22,44,45]. Therefore, we used a mouse model of early bladder carcinogenesis induced by BBN to analyze the effects of a vitamin A-rich diet on bladder urothelium. Our results demonstrate that vitamin A decreases urothelial atypia and the level of apoptosis, probably by altering the RA signaling pathway.

Normal urothelium with terminally differentiated umbrella cells was not affected by a vitamin A-rich diet. Urothelial morphology and differentiation-related urothelial markers were unaltered, as was the expression of genes involved in RA signaling (except two genes of RA metabolism, *Cyp26b1* and *Lrat*). Some studies suggest that RA signaling controls urothelial development and regeneration and plays a role in cell differentiation, proliferation, and apoptosis [19,20,36]. We analyzed the markers of urothelial differentiation, revealing that there was no change in the differentiation stage in the BBN and BBN + VitA groups. Therefore, we conclude that a vitamin A-rich diet does not alter the differentiation process of urothelial cells during early carcinogenesis. The bladder carcinogenesis process of humans and mice usually starts with cytological and architectural changes of urothelium, leading to urothelial atypia and subsequent dysplasia. Although it was previously demonstrated that vitamin A decreases bladder carcinogenesis in humans and mice, there are no data about early bladder carcinogenesis. In our study, acute BBN treatment caused widespread urothelial atypia in nearly 70% of mice and moderate atypia in the remaining 30% (Figure 1). Histopathological evaluation showed that in 19% of animals in the BBN + VitA group, urothelium remained normal or exhibited only focal atypia (Figure 1). This is in agreement with studies showing the protective effects of vitamin A [21,22,44,45]. On the other hand, the percentage of animals with desquamated urothelium did not differ between BBN and BBN + VitA groups. A diet rich in vitamin A did not have any effect on the proliferation of urothelial cells in the normal urothelium (no difference between NT and VitA groups) and in early bladder carcinogenesis (no difference between BBN and BBN + VitA groups) (Figure 4). Apoptosis, on the contrary, was decreased by a vitamin A rich diet (statistically significant difference between BBN and BBN + VitA groups) (Figure 4). Although apoptosis is mainly inhibited in cancer cells, it was shown that in early carcinogenesis, apoptotic cell death is increased in the urinary bladder [17,18]. We therefore assume that in our study decreased apoptosis due to vitamin A had a protective effect against early bladder carcinogenesis (see below). This effect could be due to an RA-induced decrease of mitochondrial oxygen species production and improved antioxidant capacity [46]. It seems that vitamin A has a protective role against early bladder carcinogenesis by decreasing atypia and apoptosis of urothelial cells, not by changing proliferation or influencing the differentiation of urothelial cells.

To illuminate the molecular background of the observed effects of a vitamin A rich diet, we measured the expression levels of key genes involved in urothelial differentiation and RA signaling. In the normal urothelium (NT and VitA groups), *Upk1a* and *Upk1b* gene expression was significantly upregulated because of vitamin A, while the expression of other uroplakin genes (*Upk2, Upk3a, Upk3b*) was unaltered (Figure 3). *Hoxb4*, which was reported to be involved in urothelial differentiation [47], was upregulated in the VitA group compared to the NT group. We therefore assume that *Hoxb4* might induce the expression of *Upk1a* and *Upk1b* in the normal urothelium. In the BBN group as well as in the BBN + VitA group all *Upks* were downregulated, which is in accordance with the studies showing that urothelial carcinogenesis is accompanied by dysregulation of *Upks* expression [10,11,12,13]. *Pparg* and *Rxra* were also downregulated in the BBN and the BBN + VitA group (Figure 5). Therefore, *Pparg/Rxra* downregulation could be responsible for *Upks* downregulation resulting in lower urothelial differentiation during early bladder carcinogenesis. Likewise, it was shown by others that the expression of *Pparg* and *Rxra* leads to partial urothelial differentiation [48,49,50].

*Pparg* expression was also linked to the urobasal subtype of bladder cancer, i.e., low-stage Ta tumors [47,50]. Additionally, Ericsson et al. showed that *Pparg*, *Rxra*, and *Foxa1* expression was maintained in a majority of urobasal tumors, therefore it is assumed that this likely contributes to the differentiated phenotype seen in this tumor subtype [47]. By comparing BBN and NT groups, we show that in early bladder carcinogenesis, *Foxa1*, *Hoxa5*, *Ppara*, *Pparg*, and *Rxra* were downregulated (Table 1, Figure 5). This might contribute to the lower differentiation state of urothelial cells. On the other hand, comparing the BBN + VitA group to the NT group, no significant downregulation was detected (except for Foxa1) (Table 1, Figure 5). The expression of *Wnt* target *Sox9* was upregulated in early bladder carcinogenesis (BBN group compared to NT group), which is in accordance with studies showing upregulation of *Sox9* in bladder cancer [51,52]. Surprisingly, *Wnt5a* was downregulated in the BBN group compared to the NT group. Nevertheless, comparing the BBN + VitA group to the NT group, *Sox9* and *Wnt5a* were not significantly altered (Figure 5). These results might be correlated with decreased urothelial atypia. To sum up, we hypothesize that a vitamin A rich diet exerts a protective role by neutralizing not only *Pparg*, *Ppara*, *Rxra*, and *Hoxa5* but also WNT5A/SOX9 signaling crosstalk.

Retinol-binding protein (RBP), which mediates retinol transport in blood plasma, and STRA6, which is an RA receptor and translocator across the plasma membrane, are proposed as potent protooncogenes and key players in cancer stem cell maintenance [53,54]. A comparison of BBN and NT groups showed that *Rbp4* and *Stra6* genes were down- and upregulated, respectively (Figure 5). On the contrary, comparing the BBN + VitA and NT groups, we documented that *Rbp4* and *Stra6* expression did not change. We also show that vitamin A decreased urothelial atypia and therefore can speculate that vitamin A exerts its protective effects by neutralizing *Stra6* and *Rbp4* expression, leading to diminished cancer stem cell preservation. We document that in the BBN + VitA group, compared with the BBN group, the apoptotic index was decreased (Figure 4). Since STRA6 induced apoptosis in urothelial cancer cells in vitro, we assume that decreased apoptosis correlates with normalized *Stra6* expression achieved by a vitamin A rich diet. An in vitro study by Carrera et al. revealed that STRA6 was present in the plasma membrane and the cytosol of the urothelial cancer cells [55]. Our results confirm this finding in an in vivo animal model of early bladder carcinogenesis. Additionally, we show that in normal urothelial cells of the NT and VitA groups, STRA6 was present in the cytoplasm. During early bladder carcinogenesis in the BBN and BBN + VitA groups, cytoplasmic localization of STRA6 was unaltered.

Previously it was demonstrated that *Lrat* downregulation accompanies different cancers, such as prostate, renal, breast, and bladder cancer [32,33,34]. A vitamin A rich diet in early bladder carcinogenesis (BBN + VitA group) caused *Lrat* upregulation, which is in accordance with *Lrat* regulation by exogenous retinoids [55]. Boorjian et al. demonstrated that LRAT immunolabeling was stronger in superficial bladder tumors than in invasive bladder tumors [35]. Our results support this notion and show that even normal urothelium is LRAT positive. In normal urothelium (NT and VitA groups), LRAT was localized in the cytoplasm of urothelial cells. However, we demonstrate that during early bladder carcinogenesis in the BBN and BBN + VitA groups, LRAT was translocated from the cytoplasm to the nuclei of urothelial cells. These results are in accordance with the report of Simmons et al. showing that LRAT could have nuclear localization [56]. LRAT has a region of 12 amino acids that harbor DNA binding properties [56,57]. Simmons et al. speculated that isoforms of LRAT may exist that possess tumor-suppressing properties. Our results support this notion, since upregulated *Lrat* in the BBN + VitA group could have contributed to decreased urothelial atypia due to vitamin A. Additional support for this is that the only other gene that was altered during BBN and vitamin A treatment compared to BBN treatment alone was the gene for NEUROD1, while the expression of *Hoxa5*, *Ppara*, *Pparg*, *Rxra*, *Sox9*, and *Wnt5a* was normalized. Retinoic acid was recently proven to induce *NeuroD1* overexpression in stem cells in vitro [58], which is in agreement with our in vivo results. One of the biological functions of NEUROD1 is nucleocytoplasmic transport [59]. Therefore, it is possible that NEUROD1 is responsible for LRAT transport from the cytoplasm into the nucleus during early bladder carcinogenesis.

## 4. Materials and Methods

### 4.1. Reagents

BBN was obtained from TCI Europe N.V., standard chow diet from Mucedola (Milan, Italy), and vitamin A rich diet from Altromin (Lage, Germany). The ClinRep^®^ HPLC Complete Kit for Vitamins A and E for measurement in plasma and analytical column with test chromatogram were from Recipe (Munich, Germany). All primary and secondary antibodies used are listed in Table 2. 3,3′-Diaminobenzidine (DAB) and avidin-biotin complex conjugated with horseradish peroxidase (ABC/HRP) were purchased from Dako and Vectashield, with 4’,6-diamidino-2-phenylindole (DAPI) from Vector Laboratories. QIAzol was obtained from Qiagen, high-capacity cDNA reverse transcription kit and Power SYBR Green master mix from Applied Biosystems, and RT^2^ Profiler PCR Array for mouse retinoic acid signaling (PAMM-180Z) from SABiosciences. All other reagents were from Merck KGaA (Darmstadt, Germany).

### 4.2. Animals

Eighty-four adult C57BL/6 male mice were used in this study. All experiments were approved by the Ethics Committee of the University of Split School of Medicine (permit no. 2181-198-03-04-14-0036) and were carried out in compliance with the ARRIVE guidelines and in accordance with the UK Animals (Scientific Procedures) Act, 1986, and associated guidelines, and EU Directive 2010/63/EU for animal experiments. Mice were housed in plastic cages at 23 ± 2 °C and 50–60% relative humidity under a 12 h·light/dark cycle.

Mice were divided into four groups by simple random sampling (Figure 1A). The NT (not treated) group had a standard chow diet containing 16,000 IU of vitamin A (Mucedola, Italy) and tap water available ad libitum. The VitA group had a vitamin A rich diet with 36.6-fold excess vitamin A for 3 weeks (586.081 IU of vitamin A as retinyl acetate; Altromin International, Lage, Germany), which was previously demonstrated to be nontoxic to mice [43], and tap water available ad libitum. The BBN group had a standard chow diet and 0.05% BBN in tap water available ad libitum for 2 weeks. The BBN + VitA group had 0.05% BBN in tap water available ad libitum for 2 weeks and a vitamin A rich diet available ad libitum 1 week prior to BBN and 2 weeks together with BBN. During the experiments, mice and the food and water they consumed were weighted every second day, and no differences were observed between the groups of animals.

Three weeks after the beginning of treatment, mice were anaesthetized by a mixture of ketamine (100 mg per kg of body weight) and xylazine (10 mg per kg of body weight). Blood was collected in test tubes without anticoagulants by using injection without a needle (to avoid erythrocyte lysis) into the abdominal vein. Tubes with blood were left at room temperature for 20 min for coagulation to occur and then centrifuged for 5 min at 2000 *g* at 4 °C. Serum was then separated from the pellet and frozen at −20 °C. After blood collection, mice were sacrificed, and their bladders were aseptically removed and processed for RNA isolation and microscopy.

### 4.3. Serum Vitamin A Measurement

For quantitative analysis of vitamin A in the collected mouse samples, a commercial ClinRep^®^ HPLC Complete Kit (Recipe Chemicals, Munich, Germany) and HPLC system (Thermo Fisher Scientific Inc., Waltham, MA, 02451, USA) were used. All necessary chemicals and an analytical chromatographic column were delivered with the kit. Amounts in the original sample preparation procedure (150 µL) were proportionally reduced (due to mouse size) as follows: 100 µL of sample (calibrator, control, or mouse serum) was transferred into a sample preparation vial. Then 100 µL of Precipitant P (containing 0.6 µg internal standard) was added and mixed on a vortex mixer for 30 s. Afterwards, samples were centrifuged for 5 min at 10,000 *g*, then 80 µL of supernatant was pipetted into a reaction vial and 80 µL of cooled (4 °C) stabilizing reagent (S) was added. Then samples were briefly mixed and centrifuged for 5 min at 10,000 *g*, and 50 µL of supernatant was injected into HPLC system with UV detector (Spectra system UV 1000, Thermo Fisher Scientific Inc.). The analytical column and mobile phase from the kit were used to separate analyte and internal standard. The flow rate of delivered mobile phase was 1.5 mL/min and column temperature was 30 °C. The initial wavelength of the detector was 325 nm, and it switched to 295 nm after 3.5 min. The chromatographic run was 8 min long, retention time of vitamin A was approximately 1.9 min, and internal standard was approximately 4.9 min. For the calibration curve in the range of 0.035–17.45 µmol/L, lyophilized serum calibrator was used. All measurements of mouse samples were performed on the same day, and analytical data originate from a single injection. For quality control of analytical determinations, ClinChek^®^ serum controls were used. Calculations were performed according to the instruction manual (ClinRep^®^ HPLC Complete Kit for Vitamins A and E, Recipe, Munich, Germany). According to the instructions, interassay precision is from 4.88 to 5.13%. All parameters were in the range of the defined criteria of the kit.

Data were analyzed with ChromQuest software (Thermo Fisher Scientific Inc.) and presented as mean ± SE. Statistical analysis (ANOVA, *F*-test, two-sided Student’s *t*-test) was performed with Excel 2016.

### 4.4. Characterization of Urothelial Lesions

The bladder samples were fixed in 4% formaldehyde in phosphate-buffered saline (PBS) and embedded in paraffin. Paraffin sections were stained with hematoxylin and eosin. Two independent pathologists experienced in urological pathology who were blinded to the mouse treatment performed histological examination two times. Desquamation of urothelium was defined as urothelial cells detached from lamina propria and was observed in the lumen of the bladder. Histomorphological changes of the urothelium indicating early bladder carcinogenesis were diagnosed when the presence of one or more cellular or architectural features that deviated from an otherwise normal-appearing urothelium were observed, i.e., abnormal nuclear features, including increased size (nucleomegaly), deviation from typical ovoid shape, coarse chromatin, irregular nuclear membrane contour, prominent nucleolus or multiple nucleoli, cellular crowding, and mild hyperchromasia. Since morphological changes were mild, with no cases showing features meeting the criteria for urothelial carcinoma in situ (CIS), they were not subclassified further and were named simply atypia.

### 4.5. Immunohistochemistry

Sections were deparaffinized in xylene, hydrated by descending concentrations of ethanol, and microwave heated (3 × 5 min at 600 W) in TRIS/EDTA buffer (pH 9.0). Endogenous peroxidase activity was blocked with 3% H_2_O_2_ in methanol and nonspecific labelling was blocked by 5% bovine serum albumin (BSA) in PBS buffer. Sections were incubated overnight at 4 °C with anti-cleaved caspase-3 or anti-uroplakins (Table 2), diluted in 1% BSA in PBS. For negative controls, incubation with primary antibody was omitted or the specific primary antibody was replaced by a nonrelevant antibody. Secondary antibodies (Table 2) were diluted in PBS and applied to the sections for 1 h at room temperature. Then 30 min incubation with ABC/HRP complex was performed according to the manufacturer’s instructions. After washing in PBS, the standard DAB development procedure was performed and sections were thoroughly washed in water. Sections were then counterstained with hematoxylin and examined with an Olympus BX43 microscope (Olympus Corporation, Tokyo, Japan).

### 4.6. Immunofluorescence

Sections were deparaffinized in xylene, hydrated by descending concentrations of ethanol, and microwave heated in TRIS/EDTA buffer (pH 9.0). Then sections were preincubated with 5% BSA in PBS for 2 h at room temperature to block nonspecific labelling. Sections were incubated at 4 °C overnight with primary antibodies: anti-KRT20 (keratin20), anti-Ki-67, anti-STRA6, and anti-LRAT (Table 2), diluted in 1% BSA in PBS. For negative controls, incubation with primary antibodies was omitted. After washing, sections were incubated overnight at 4 °C with proper secondary antibodies (Table 2) diluted in PBS. Then, sections were thoroughly washed and mounted in Vectashield with DAPI. Sections were examined and imaged with an AxioImager Z1 fluorescence microscope (Zeiss, Oberkochen, Germany).

### 4.7. Scanning Electron Microscopy

To evaluate the differentiation state of urothelial cells, changes of the apical plasma membrane structure were analyzed by scanning electron microscopy. Bladder samples were fixed in 4.5% formaldehyde and 2% glutaraldehyde for 3 h. After washing, the samples were postfixed in osmium tetroxide for 1 h, washed, and dehydrated. Then critical-point drying was performed with liquid CO**_2_**. Dry samples were fixed to the appropriate holder. After gold sputter-coating, they were examined at 30 kV with a Tescan Vega3 scanning electron microscope (Tescan, Brno, Czech Republic). 

### 4.8. PCR Array and qPCR Experiments

RNA was extracted using QIAzol (Qiagen) from bladder tissue according to the manufacturer’s instructions. In the final step, RNA was dissolved in water and 1 μg of total RNA was used for preparing cDNA. The high-capacity cDNA reverse transcription kit was used for cDNA preparation according to the manufacturer’s instructions. An equal amount of RNA from three mice were pooled for cDNA preparation. Real-time PCR was carried out with Power SYBR Green master mix using a 7500 Real-Time PCR System (Applied Biosystems, now Thermo Fisher Scientific Inc., Waltham, MA, 02451, USA). Actin level of expression was used as endogenous control. All primers used are listed in Table 3. An RT^2^ Profiler PCR Array for mouse retinoic acid signaling (Qiagen, Germantown, MD, 20874, USA) was used to analyze the retinoic acid signaling pathway according to the manufacturer’s instructions. Three plates with a total of nine mice were analyzed per treatment group.

### 4.9. Statistical Analysis of Urothelial Changes and Bioinformatics

Urothelial lesions determined by pathologists were first expressed as percentage of desquamated urothelium and urothelial atypia for each animal, and these percentages were divided into three groups: focal occurrence (0–30%), moderate occurrence (31–70%), and widespread occurrence (71–100%). Then the results were represented as percentage of animals with focal, moderate, and widespread desquamation or atypia. Statistically significant differences between BBN and BBN + VitA groups were tested by ANOVA, F-test, and two-sided Student’s *t*-test.

As a marker of urothelial differentiation, the presence of microvilli, ropy ridges, and urothelial plaques was analyzed by scanning electron microscopy, and the presence of positive KRT20 and uroplakin immunolabeling in superficial, intermediate, and basal urothelial cells was analyzed by light microscopy. All data were evaluated by a four-point scale (0, none; 1, mild; 2, moderate; 3, strong presence) [12]. 

The apoptotic and proliferation index was determined by immunolabeling against cleaved caspase-3 and Ki-67, respectively. Paraffin sections of whole bladder were made for animals from each group (NT, VitA, BBN, BBN + VitA). The number of urothelial cells counted in each group of animals was 3000–8500. The apoptotic and proliferation index was defined as percentage ratio of positively labelled cells to total number of counted cells. The values were expressed as mean ± SE. Differences between the means of various groups were tested by ANOVA, *F*-test, and two-sided Student’s *t*-test.

The data from PCR were statistically analyzed using GraphPad Prism 8. Two-sided Student’s *t*-test was used to assess differences in transcript expression of genes analyzed with RT2 profiler plates. Actin expression level was used as endogenous control for normalization. The ΔΔCT comparative method was used for analysis, and data are expressed as fold change ratio. Data are expressed as Log2 fold change (FC) in heatmap and volcano plots using GraphPad Prism 8. ANOVA followed by Tukey’s post hoc test was used for analysis of uroplakin gene expression.

For each experiment, the number of biological replicates (animals used) is stated in the figure captions as “*n* =” while the number of technical replicates was one for histopathological evaluation, immunolabeling, and scanning electron microscopy and two for qPCR experiments.

## 5. Conclusions

The emerging picture is that a vitamin A rich diet diminishes early bladder carcinogenesis by decreasing urothelial atypia and apoptosis of urothelial cells. The decreased atypia is probably due to diminished up- or downregulation of genes for transcription factors (*Pparg*, *Ppara*, *Rxra*, *Hoxa5*, and *Sox9*) and RA pathway regulators (*Wnt5a*). During early bladder carcinogenesis, apoptosis of urothelial cells is increased, probably because of upregulated expression of *Stra6*. A vitamin A rich diet results in diminished apoptosis of urothelial cells, which might be correlated to the neutralized expression of *Stra6*. The only genes that are upregulated during early bladder carcinogenesis with a vitamin A rich diet compared to without a vitamin A rich diet are *Lrat* and *Neurod1*. We show that during early bladder carcinogenesis, LRAT is translocated from the cytoplasm of urothelial cells to the nuclei. In conclusion, a vitamin A-rich diet exerts a protective effect during the early phases of urinary bladder carcinogenesis, most probably by regulating the pattern of gene expression and influencing the level of apoptosis of urothelial cells.

## Figures and Tables

**Figure 1 cancers-12-01712-f001:**
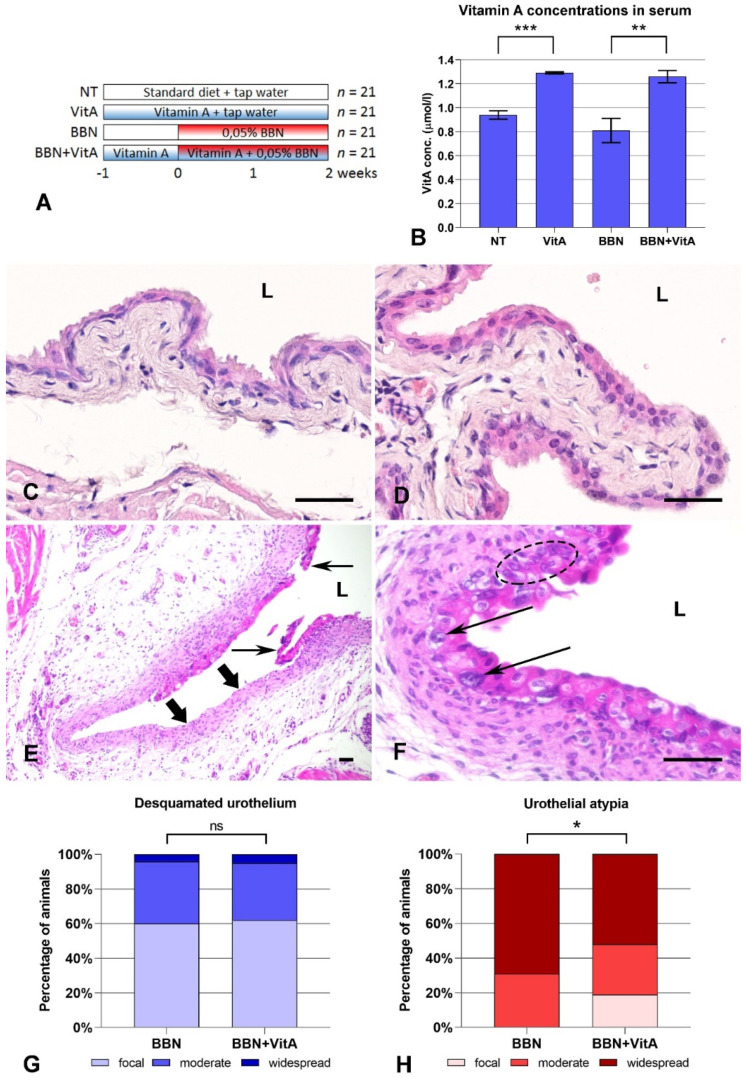
Treatment regiments, vitamin A serum concentrations, and histopathological changes in urothelium after *N*-butyl-*N*-(4-hydroxybutyl) nitrosamine (BBN) treatment and vitamin A rich diet: (**A**) Four groups of animals treated with standard diet (NT), vitamin A rich diet (VitA), BBN in drinking water (BBN), and BBN in drinking water and vitamin A rich diet (BBN + VitA); (**B**) vitamin A serum concentrations of NT (*n* = 6), VitA (*n* = 6), BBN (*n* = 9), and BBN + VitA (*n* = 9) groups of animals (tested by ANOVA, *F*-test, and two-sided Student’s *t*-test; *** *p* = 0.0001, ** *p* = 0.0011). Hematoxylin and eosin staining: Normal urothelium of (**C**) NT group and (**D**) VitA group showing characteristic three-layered urothelium with superficial umbrella cells; (**E**) desquamated urothelium (thin arrows) in BBN group with denuded basal lamina and lamina propria (thick arrows). Similar areas were observed in BBN + VitA group; (**F**) urothelium of BBN + VitA group with large pleomorphic and hyperchromatic nuclei with conspicuous nucleoli and irregular contours (thin arrows), with nuclear crowding (dashed ellipse) and loss of cell polarity, characteristic of urothelial atypia. Similar areas, but even more widespread, were observed in BBN group. L, lumen of the bladder. Scale bars = 50 μm. Histopathological evaluation of (**G**) desquamated urothelium and (**H**) urothelial atypia in BBN (*n* = 21) and BBN + VitA (*n* = 21) groups (tested by ANOVA, *F*-test, and two-sided Student’s *t*-test; * *p* < 0.05; ns, no statistically significant difference). In NT (*n* = 21) and VitA (*n* = 21) groups, the percentage of animals with desquamation of urothelium or urothelial atypia was zero.

**Figure 2 cancers-12-01712-f002:**
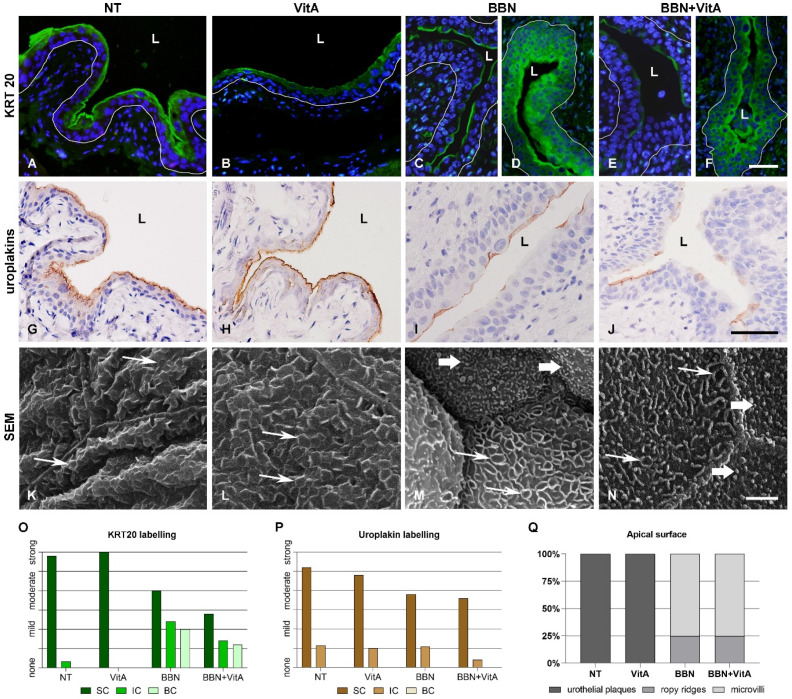
Differentiation markers of urothelium after BBN treatment and vitamin A rich diet. (**A**–**D**) Keratin 20 (KRT20) immunofluorescence. KRT20 labelling (green) is strong in the umbrella cells of NT and VitA groups. In BBN and BBN + VitA groups, two different areas are observed. (**C**,**E**) In some areas, the apical cytoplasm of individual superficial cells is positive and (**D**,**F**) in other areas, superficial, intermediate, and even basal cells are positive. White line depicts the location of basal lamina. L, lumen. Scale bars = 50 μm. Uroplakin immunohistochemistry: (**G**,**H**) Uroplakin labelling (brown) is prominent in all superficial urothelial cells of NT and VitA groups; (**I**,**J**) only some superficial urothelial cells of BBN and BBN + VitA groups are uroplakin positive. Scale bars = 50 μm. Scanning electron microscopy: Apical plasma membrane of umbrella cells shows (**K**,**L**) urothelial plaques (thin arrows) in NT and VitA groups and (**M**,**N**) ropy ridges (thin arrows) and microvilli (thick arrows) in BBN and BBN + VitA groups. Scale bars = 5 μm. Four-point scale evaluation of differentiation markers in NT (*n* = 6), VitA (*n* = 6), BBN (*n* = 12), and BBN + VitA (*n* = 12) groups: (**O**,**P**) KRT20 and uroplakin labelling as observed in superficial (SC), intermediate (IC), and basal (BC) urothelial cells; (**Q**) urothelial apical surface as observed by scanning electron microscope showing percentage of apical surface covered by urothelial plaques, ropy ridges, and microvilli.

**Figure 3 cancers-12-01712-f003:**
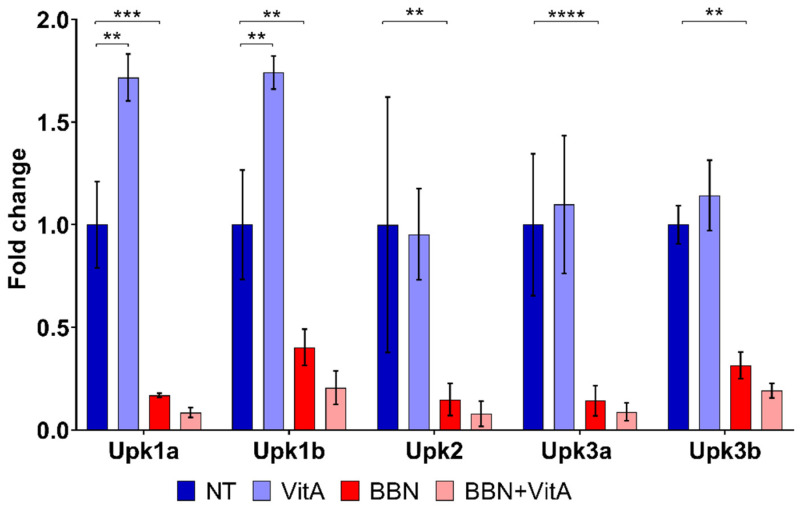
Transcriptional expression profiles of uroplakin genes (*Upk1a*, *Upk1b*, *Upk2*, *Upk3a*, and *Upk3b*) in the bladder wall of four groups of animals: NT (*n* = 9), VitA (*n* = 6), BBN (*n* = 9), BBN + VitA (*n* = 9). Statistical analysis was performed using ANOVA followed by Tukey’s post hoc test; ** *p* < 0.005, *** *p* < 0.0005, **** *p* < 0.0001.

**Figure 4 cancers-12-01712-f004:**
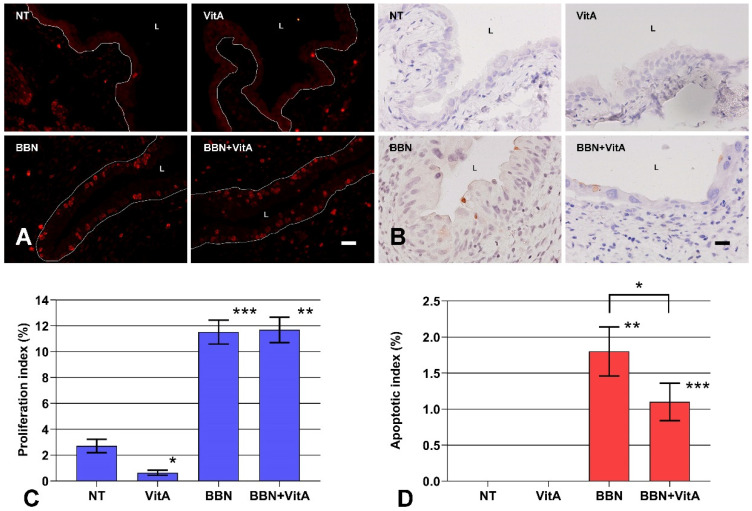
Proliferation and apoptosis of urothelial cells. (**A**) Ki-67 immunofluorescence (red) for identification of proliferating cells. Red nuclei are Ki-67 positive nuclei from proliferating cells; white line depicts the location of basal lamina. L, lumen. Scale bar = 50 μm. (**B**) Active caspase-3 immunohistochemistry (brown) for identification of apoptotic cells. Brown cells are active caspase-3 positive cells dying via apoptosis. Scale bar = 50 μm. (**C**) Proliferation indices of urothelial cells in NT (*n* = 6), VitA (*n* = 6), BBN (*n* = 12), BBN + VitA (*n* = 12). Statistically significant differences (tested by ANOVA, F-test, and two-sided Student’s t test) were determined between NT and VitA (* *p* = 0.0062) and between NT and BBN (*** *p* = 2.22 × 10^−9^) and VitA and BBN + VitA (** *p* = 3.98 × 10^−9^). There was no statistically significant difference between BBN and BBN + VitA; (**D**) apoptotic index of urothelial cells in NT (*n* = 6), VitA (*n* = 6), BBN (*n* = 12), BBN + VitA (*n* = 12). Statistically significant differences (tested by ANOVA, *F*-test, and two-sided Student’s *t* test) were determined between NT and BBN (** *p* = 0.0032) and VitA and BBN + VitA (*** *p* = 0.00047), and between BBN and BBN + VitA (* *p* = 0.049).

**Figure 5 cancers-12-01712-f005:**
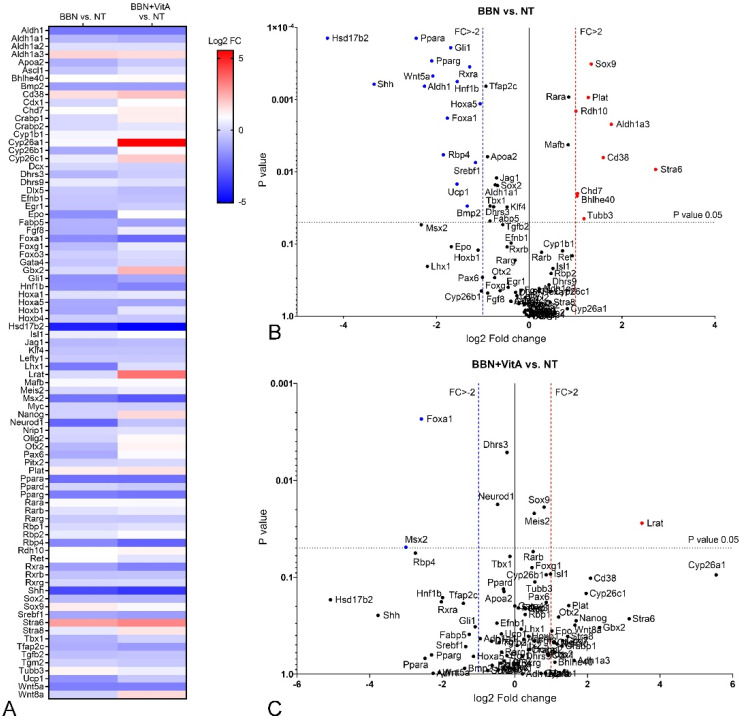
Differential expression of genes involved in retinoic acid signaling in mouse bladder tissue following BBN treatment and vitamin A rich diet. Comparisons of NT (*n* = 9), BBN (*n* = 9), and BBN + VitA (*n* = 9) groups of animals. (**A**) Heatmap representation of fold change in gene expression in BBN vs. NT and BBN + VitA vs. NT. (**B**,**C**) Volcano plots showing comparison of BBN vs. NT and BBN + VitA vs. NT. For each group, three cDNAs, each containing RNA from three different mice, were used. Significantly affected gene expression with fold change greater then 2, upregulated or downregulated, is marked in red or blue, respectively (*p* value < 0.05; tested by two-sided Student’s *t*-test). FC, fold change.

**Figure 6 cancers-12-01712-f006:**
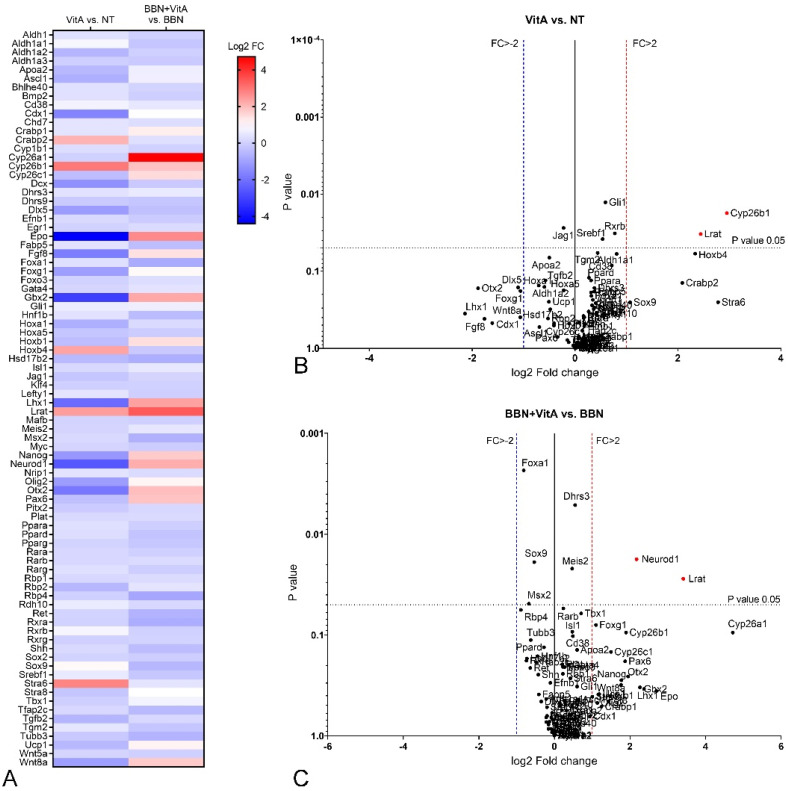
Differential expression of genes involved in retinoic acid signaling in mouse bladder tissue after BBN treatment with or without vitamin A rich diet. Comparisons of NT (*n* = 9), VitA (*n* = 9), BBN + VitA (*n* = 9), and BBN (*n* = 9) groups. (**A**) Heatmap representation of fold change in gene expression in VitA vs. NT and BBN + VitA vs. BBN groups. (**B**,**C**) Volcano plots showing comparison of VitA vs. NT and BBN + VitA vs. BBN groups. For each group, three cDNAs, each containing RNA from three different mice, were used. Significantly affected gene expression with fold change greater then 2, upregulated or downregulated, is marked in red or blue, respectively (*p* value < 0.05; tested by two-sided Student’s *t* test). FC, fold change.

**Figure 7 cancers-12-01712-f007:**
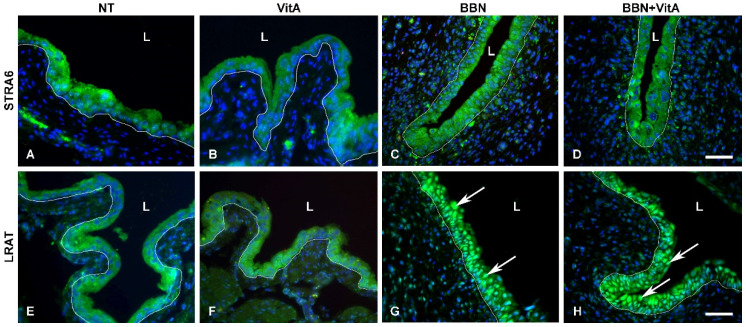
Immunofluorescence of proteins stimulated by retinoic acid 6 (STRA6) and lecithin retinol acyltransferase (LRAT). (**A**–**D**) STRA6 labelling (green) is strong in the cytoplasm of urothelial cells of all groups. Scale bar = 50 μm. LRAT labelling (green) is present in the cytoplasm of the normal urothelial in (**E**) NT and (**F**) VitA groups. In (**G**) BBN and (**H**) BBN + VitA groups, LRAT labelling is strongest in the nuclei of urothelial cells (arrows). White line depicts the location of basal lamina, L, lumen. Scale bar = 50 μm.

**Table 1 cancers-12-01712-t001:** Differential expression of the genes involved in retinoic acid (RA) signaling in the mouse bladder tissue following BBN treatment and vitamin A rich diet. The list of top upregulated and downregulated genes from volcano plots with fold change higher than 2 or lower than −2 and with *p* < 0.05 (comparisons between BBN and NT, BBN + VitA and NT, VitA and NT, and BBN + VitA and BBN are shown).

Differential Expression	Gene	Fold Change	*p*-Value	Gene	Fold Change	*p*-Value
	BBN vs. NT.			VitA vs. NT		
**Upregulated**	*Stra6*	6.61	0.0092	*Cyp26b1*	7.72	0.0176
	*Adh1a3*	3.41	0.0022	*Lrat*	5.42	0.0329
	*Cd38*	3.02	0.0063			
	*Sox9*	2.53	0.0003	**BBN + VitA vs. BBN**		
	*Plat*	2.42	0.0009	*Neurod1*	4.51	0.0177
	*Tubb3*	2.26	0.0445	*Lrat*	10.64	0.0276
	*Chd7*	2.06	0.0199			
	*Bhlhe40*	2.05	0.0216	**BBN + VitA vs. NT**		
	*Rdh10*	2.01	0.0014	*Lrat*	11.31	0.0276
**Downregulated**	*Hoxa5*	−2.08	0.0011	*Foxa1*	−5.96	0.0023
	*Srebf1*	−2.23	0.0074	*Msx2*	−8.02	0.0489
	*Rxra*	−2.43	0.0004			
	*Bmp2*	−2.52	0.0298			
	*Hnf1b*	−2.93	0.0006			
	*Ucp1*	−2.94	0.0148			
	*Gli1*	−3.23	0.0002			
	*Foxa1*	−3.39	0.0018			
	*Rbp4*	−3.61	0.0058			
	*Wnt5a*	−4.22	0.0005			
	*Pparg*	−4.29	0.0003			
	*Adh1*	−4.77	0.0007			
	*Ppara*	−5.41	0.0001			
	*Shh*	−10.14	0.0006			
	*Hsd17b2*	−20.35	0.0001			

**Table 2 cancers-12-01712-t002:** List of primary and secondary antibodies used for immunolabeling.

Primary Antibody	Dilution	Cat. No.	Provider
polyclonal rabbit anti-cleaved caspase-3	1:200	9661	Cell Signaling Technology^®^
polyclonal rabbit anti-uroplakins (UPs) ^1^	1:10,000	–	Prof. Tung-Tien Sun
monoclonal mouse anti-KRT20	1:200	M7019	Dako
polyclonal rabbit anti-Ki-67	1:200	ab15580	Abcam
polyclonal rabbit anti-STRA6	1:500	bs-12351R	Bioss Antibodies
polyclonal rabbit anti-LRAT	1:50	PA5-38556	Thermo Fisher Scientific
**Secondary Antibody**			
anti-rabbit HRP	1:500	P0448	Dako
anti-rabbit Alexa Fluor 555	1:400	A21428	Thermo Fisher Scientific
anti-rabbit Alexa Fluor 488	1:400	A11008	Invitrogen
anti-mouse Alexa Fluor 488	1:400	A11001	Thermo Fisher Scientific

^1^ These antibodies react strongly against UPIIIa and weakly against UPIa, UPIb, and UPII.

**Table 3 cancers-12-01712-t003:** List of primers used for real-time PCR.

Gene	Gene ID	Primer Sequence
Upk1a	NM_026815	F 5'TTTGGTGTAGGAGCTGCG3'
		R 5'GTTATCAGGGATGGGTTGGAC3'
Upk1b	NM_178924	F 5'GGGTGGAGAATAACGATGCTG3'
		R 5'GGTCCAGAGATCAGTTCATAGC3'
Upk2	NM_009476	F 5'GCGCATATCAGGTGACAAAC3'
		R 5'CCATTCCTAACCCAATAGACTCC3'
Upk3	NM_023478	F 5'GGAGTGGAGGCATGATTGTC3'
		R 5'GGTGATCTGTGAGTCGTGTG3'
Upk3b	NM_175309	F 5'TCCAACCCCATTTATCTCCAC3'
		R 5'GACGATCATACAGCCGCTC3'

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
