# Peer review of "Vitamin A Rich Diet Diminishes Early Urothelial Carcinogenesis by Altering Retinoic Acid Signaling"

_cancers, 2020, doi:10.3390/cancers12071712_

Round 1
Reviewer 1 Report
- This is an important area of study in carcinogenesis and of interest to readers. This manuscript is too long for the content presented.
- A major interesting result is that compared to BBN treatment, the genes Neurod1 and Lrat were identified as differentially expressed in VitA+BBN mice as seen by mRNA expression array profiling.
- In results section, many sentences are long and sometimes unclear. One example, pg 3 of 21 last line “…by decreasing BBN induced urothelial atypia” should be “… by decreasing BBN-induced urothelial atypia”. Sentence structures can be fixed.
- On what basis is reduction of apoptosis considered protective in early bladder cancer? Since this is contrary to commonly understood mechanisms of carcinogenesis or cancer progression, it would be useful to add a few sentences in the Introductions section on pg2. Currently there is none. If fluctuations in apoptosis are found, that may be mentioned further in Discussion.
- Supplementary Fig can be incorporated in Fig 1.
- Results relating to Fig 2 should state clearly the differences found between BBN-induced vs BBN+VitA groups in terms of differentiation stages, uroplakin staining, apical plasma as seen by SEM. These important observations need to be clearly stated with differences or no change ( no change in apical structure). Similarly throughout the manuscript differences or no differences found between BBN-induced vs BBN+VitA groups should be clear.
- The volcano plots and heat maps for Fig 5 and 6 maybe combined into one panel.
- pg 10: para 2, “Number of differentially expressed genes…” How many?
- There is useful data that can be showcased. From the volcano plot , one table of the top 10 significant upregulated/downregulated genes is needed. The table can have three columns: NT vs BBN, NT vs Vit A, and BBN vs BBN+VitA, genes of various fold changes and should be ranked by their log2Fold change and by P value. Of these three, NT vs VitA gene expression changes will serve as future reference for researchers from other fields as well.
- For those genes that were identified as upregulated (Neurod1, Lrat) : Validation studies was not performed by protein expression or PCR. A localization study is shown.
- Localization study in relation to nucleocytplasmic transport : Difference between the BBN-induced and BBN+vitA animals are not clear in text, though discussion section is good on pg 13
- Introduction can mention what constitutes vitA rich concentration in serum since Vit A or retinoid toxicity in humans from vitA supplements is known.
- Format: References are numbered and then appear as text in Discussion, pg 11
- Methods section: 4 animal treatment groups can be depicted as a bar diagram of treatment regimens of vit A and BBN or prevention strategy.
Reviewer 2 Report
In the manuscript “Vitamin A Rich Diet Diminishes Early Urothelial Carcinogenesis by Altering Retinoic Acid Signaling”, Zupančič et al. analyzed the effects of vitamin A (VitA) on early urothelial carcinogenesis by studying morphological and molecular features of the urothelial mucosa in mice exposed for three weeks to either the carcinogen N-butyl-N-(4-7 hydroxybutyl)nitrosamine (BBN) alone or to BBN + VitA. They found that urothelial cell desquamation, differentiation and proliferation rate were not altered when mice were fed with BBN+VitA compared to BBN-only exposure. In contrast, they claim that VitA decreased urothelial atypia and apoptosis, which are early signs of early carcinogenesis, in the early step of carcinogenesis induced by BBN. From the results of the molecular analysis, they concluded that VitA protects mice from early urothelial carcinogensis by contrasting the alterations in the retinoic acid signaling pathway induced by BBN in urothelial cells and increasing Lrat and Neurod1 expression, as well as LRAT translocation from cytoplasm to nuclei.
The topic of the manuscript is very interesting and the authors used different types of approaches to study it. However, I have several concerns regarding both the experimental system itself and how the results are presented and discussed.
As far as the experimental system is concerned, I believe that a very important datum is missing, that is the frequency of tumor formation in mice exposed to BBN-only and after exposure to BBN and Vitamin A. Does vitamin A decrease tumor formation in mice exposed to BBN?
As far as one of the most important conclusions of the manuscript is concerned, that is that VitA contrasts urothelial carcinogenesis by decreasing mucosa atypia, I think that the results of atypia analysis are not clear. In the Discussion, the authors states: “Histopathological evaluation showed that urothelium remains normal or exhibits only focal atypia in 20 % of animals, fed with vitamin A rich diet during BBN treatment.”, but in the Results, it is not written that urothelium was normal in the majority of the mice treated with BBN+VitA. From the plot shown in Fig. 1 and from the description, widespread atypia was reduced from 69% in the BBN group, to 52% in the BBN+VitA group. The reduction in widespread atypia was paralleled by the observation of focal atypia in some mice, not of a normal mucosa and the frequency of moderate atypia was the same in the two groups. The decrease in the percentage of mice showing widespread atypia in the BBN+VitA group compared to the BBN only-exposed group is really small. It is statistically significant, but I wonder whether it has a real meaning in terms of cancer prevention, also considering that the administration of VitA to BBN-exposed mice does not change the degree of desquamation, the lower differentiation state and proliferation of urothelial cells in BBN treated mice. A relationship between the decrease in tumor formation in BBN-VitA-exposed mice compared to BBN only-exposed mice could help to evaluate the meaning of the data reported above.
As far as the molecular results are concerned, I find that they are often commented in an unclear way, both in the Results and Discussion sessions (e.g. “Our study shows that BBN treatment alone results in Pparg and Rxra downregulation, which was diminished when vitamin A rich diet was used together with BBN (Fig. 5). We therefore assume that vitamin A reduces uroplakin expression during early bladder carcinogenesis by reducing Pparg and Rxra expression.”, to understand the final meaning of these sentences is really difficult). I think that a revision of the style could help in catching the meaning of the results.
Authors performed molecular analyses in less than a half of the mice of each group, how did they choose the mice? It would be interesting if the authors performed a cross-analysis between the histopathological and molecular results in each mouse.
In line with this, the authors show alterations in the expression of genes involved in retinoic acid signaling in mice exposed to BBN compared to untreated mice, changes that are significantly prevented when VitA is added to BBN. However, as I understand from the Results section, the reduction in the percentage of mice showing widespread atypia in these group of mice is small, how can the two data reconcile?
The discussion of the possible role of Wnt5a in early urothelial carcinogenesis is confused. Reading the Discussion, the message is that Wnt5a is overexpressed after BBN treatment, while the Results show that it is downregulated. Moreover, Wnt5 should positively regulate SOX9, but in this work, Wnt5a is downregulated while SOX9 is upregulated.
Some conclusions are also overstatetments “We show that in early bladder carcinogenesis Foxa1, Hoxa5, Ppara, Pparg and Rxra are downregulated (Fig. 5), which might contribute to lower differentiation state of urothelial cells. Vitamin A rich diet neutralize the expression of all of these genes (except for Foxa1) (Fig. 5), indicating, that this neutralization is the cause for decreased urothelial atypia.” Data show a correlation between the two phenomena, not a causal relationship.
In conclusion, I believe that in its present form the manuscript has too many problems that prevent an appropriate revision.
Reviewer 3 Report
The presented manuscript by Zupancic et al. aims at characterizing the role of Vitamin A on bladder cancer development using a BBN animal model. Histologically, Vitamin A rich diet reduce widespread atypia, thus may diminish benign lesions of the urinary bladder. On the molecular level, the authors demonstrate absence of deregulated genes involved in retinoic acid signaling such as Wnt5a or Pparg due to Vitamin A diet while a shift in translocation of LRAT, an enzyme producing all-trans retinyl esters was shown independently of Vitamin A.
The data are of potential interest for the scientific community as the study gain novel insights into the impact of Vitamin A on urothelial carcinogenesis highlighting associated genes and the RA pathway. Overall, the study has several strengths: (1) logical and thorough experimental design: a well conducted study with different consecutive steps including assessment from the histological, or the molecular level. (2) High quality of data presentation. (3) The data mainly support the conclusion that Vitamin A rich diet may has protective impact on urothelial carcinogenesis affecting genes and pathways involved in programmed cell death during early steps. However, some descriptions of procedures and methods misses depth of detail and statistics seems not suitably conducted for each data set. Beyond that there are several minor items that remain upon reading the manuscript.
Minor comments:
Comment 1: The authors stated that urothelial lesions were independently characterized by two experienced pathologists. Have the authors conducted the assessment blinded as such evaluations of marginally histological alterations may be still operator-dependent. Please comment on this.
Comment 2: The authors performed immunohistochemistry to assess differentiation of urothelial cells using markers such as Krt20. The authors further showed quantification of differentiation stages between the group of BBN and BBN+VitA. Please specify which cut-off for immunohistochemical staining (intensity?, percent of cells?, Remmele score?) was used to quantify expression of markers.
Comment 3: In Figure 3 the authors used a Student’s t test to statically compare mRNA expression (qPCR results) of uroplakin genes between groups greater than two. The test must not use in this setting: 1. the t test requires that the sampling distribution of t is normally distributed. Did the author calculate normal distribution in their test group? 2. In case of more than two groups an ANOVA test, for instance the Kruskal-Wallis test, must be used.
Comment 4: Please specify which calculation method was used for semi-quantification of qPCR analyses (ΔΔCT method?, what was used for normalization?).
Round 2
Reviewer 1 Report
Table 1 right top box should read BBN+VitA instead of BBN-VitA.
Reviewer 2 Report
The manuscript has been greatly improved. I still only have a couple of concerns.
1) In Response 5, the authors write: “We used the mix of cDNA samples from three mice for RT2 Profiler Mouse Retinoic Acid Signaling PCR Array”. The use of pooled cDNAs does not come out anywhere in the manuscript. In the Figure legends and in Materials & Methods, they write that molecular analysis was performed on 9 mice, but to individually analyze cDNAs prepared form 9 animals is not the same as to analyze 3 pools of 3 cDNAs each (if this is what they did).
2) The explanation of uroplakin downregulation in mice exposed to BBM and vitamin A it is still unclear to me. In the manuscript version 2, the authors write: “Our study shows that in the BBN group Pparg and Rxra were downregulated compared to the NT group. Yet there was no Pparg and Rxra downregulation when comparing the BBN+VitA and NT groups (Figure 5). We therefore assume that vitamin A rich diet taken together with BBN treatment reduces uroplakin expression during early bladder carcinogenesis by reducing Pparg and Rxra expression.”
If there is no no Pparg and Rxra downregulation in the BBN+VitA group compared to the NT group, how can vitamin A, when taken together with BBN, reduce uroplakin expression during early bladder carcinogenesis by reducing Pparg and Rxra expression?
